# Preparation, Structure and Properties of Acid Aqueous Solution Plasticized Thermoplastic Chitosan

**DOI:** 10.3390/polym11050818

**Published:** 2019-05-07

**Authors:** Yu Zhang, Biao-Lan Liu, Liang-Jie Wang, Ying-Hua Deng, Shi-Yi Zhou, Ji-Wen Feng

**Affiliations:** 1Hubei Key Laboratory of Purification and Application of Plant Anti-Cancer Active Ingredients, School of Chemistry and Life Sciences, Hubei University of Education, Wuhan 430205, China; starpeer@hue.edu.cn (Y.Z.); wangliangjie@hue.edu.cn (L.-J.W.); dengyinghua@hue.edu.cn (Y.-H.D.); 163zsy1998@163.com (S.-Y.Z.); 2Institute of Materials Research and Engineering (IMRE), Hubei University of Education, Wuhan 430205, China; 3Hubei Engineering Technology Research Center of Environmental Purification Materials, Hubei University of Education, Wuhan 430205, China; 4State Key Laboratory of Magnetic Resonance and Atomic Molecular Physics, Wuhan Center for Magnetic Resonance, Wuhan Institute of Physics and Mathematics, Chinese Academy of Sciences, Wuhan 430071, China; biaolanliu@foxmail.com

**Keywords:** chitosan, thermoplastic process, plasticizer, protonation

## Abstract

This work provides a simple method for the preparation of thermoplastic chitosan using the most common dilute inorganic and organic acids in aqueous solutions, namely hydrochloric acid (HCl) and acetic acid (HAc). The melting plasticization behavior of chitosan under different concentrations and types of acid solution was investigated. By means of infrared spectra (IR), scanning electron microscope (SEM), X-ray diffraction (XRD), and other characterization methods, as well as a mechanical property test, it was found that as the acid solution concentration increased, the protonation effect was stronger and the plasticization performance showed a better trend. The structure and performance of the modified chitosan were optimal when the concentration of HCl was around 8 wt %. In addition, it was found that HCl had a better effect on the plasticization of chitosan than HAc, which was because the protonation ability of HCl was stronger than that of HAc. Unlike the casting method, the structure and properties of chitosan sheets prepared by thermoplastic processing were directly affected by protonation, however not by the interaction of anionic-cationic electrostatic attractions between the –NH^3+^ groups of chitosan chains and the carboxyl groups of acetic acids or the chloridoid groups of hydrochloric acid.

## 1. Introduction

With the increasing development of synthetic polymer materials, environmental pollution problems have become increasingly prominent due to their non-biodegradability after using [1]. Fortunately, chitosan (CS), a polysaccharide, offers several advantages for the replacement of synthetic polymers in the plastic industry due to its low cost, non-toxicity, biodegradability, and availability [2,3]. Chitosan is produced by the deacetylation of chitin, which is a naturally abundant polymer material, second only to cellulose in natural resources [4]. In addition, chitosan contains amino groups and is the only nitrogen-containing alkaline polysaccharide in natural polysaccharides. Therefore, it can be used in several areas such as the food industry, waste water treatment, cosmetics, and biomedical applications due to its efficiency against bacteria and viruses and its structural characteristics [5,6,7]. In addition, chitosan exhibits excellent film-forming properties and chitosan films have been used for protection against contamination and improving food quality and shelf life [8,9].

At present, two general methods that are used to produce films are the wet and dry methods. The former is so called solvent casting, which is based on the dissolution of chitosan in dilute acid solution and the subsequent solvent evaporation to produce plasticized chitosan films [10]. Chitosan films produced with the solvent casting method are fragile at small thicknesses. On the other hand, for larger thicknesses, the internal tension increases and a rough surface is usually obtained [11]. In addition, such a method is not adequate to scale-up from a laboratory to an industrial level because it is a very time-consuming process [12]. The latter method, namely thermo-mechanical mixing, is expected to overcome these shortcomings and thus to provide an ideal process technique to fabricate chitosan-based films. This technique has been successfully applied in the preparation of different plasticized polysaccharides (i.e., thermoplastic polysaccharides) such as starch [13,14] and sodium alginate [15]. However, like cellulose, chitosan exhibits a degradation temperature below its melting point. Therefore, chitosan-based materials cannot be prepared in typically thermoplastic processing equipment incorporating solid chitosan particles. Although the plasticized chitosan can be obtained by destruction of the granules in the presence of plasticizers such as glycerol, glycol, or polyethylene glycol, the productivity level is still low.

As far as the present situation is concerned, it is of great significance to find an effective method to prepare thermoplastic chitosan. Although chitosan is insoluble in water, alkali, and most common organic solvents, fortunately it is soluble in most acid aqueous solutions [16,17]. Hence, it is possible to prepare thermoplastic chitosan using an acid solution as a plasticizer.

In recent years, there have been several papers which have reported on the preparation of thermoplastic chitosan [18,19,20,21,22]. Avérous et al. reported the preparation of plasticized chitosan by thermo-mechanical treatment with the action of an acetic acid aqueous solution and glycerol [18]. In addition, the effect of different polyol plasticizers (glycerol, xylitol, and sorbitol) on the structure and properties of thermoplastic chitosan were also investigated [12]. Another report investigated the plasticization of chitosan by thermo-compression molding with the action of a choline chloride (ChCl)-based plasticizer [22]. The above studies prove the practicability of chitosan thermoplastic processing and are fundamental in the thermoplastic chitosan preparation. In the current process method, both acid aqueous solutions as hydrogen bond donors and polyol as plasticizers were used. Although polyols can increase the processability of chitosan, they can also obviously increase the hydrophilicity of the chitosan, which limits the application of the obtained materials. Acid solutions can destroy the intra hydrogen bond of chitosan, and thus reduce the melting temperature. Therefore, acid solutions play a more important role in the processing process than polyol. However, whether chitosan can be thermoplastically processed under the action of an acid aqueous solution alone and how the chitosan interacts with the acid solution have not been studied yet.

In this work, we developed a new formulation for chitosan thermomechanical plasticization in which an acid aqueous solution was applied as both a plasticizer and a protonation agent for chitosan. Specifically, the most common and simplest inorganic acid, hydrochloric acid (HCl), and organic acid, acetic acid (HAc), solutions of different concentrations were used to prepare the thermoplastic chitosan. In order to analyze the relationships among the formulations, microstructure, and properties of different plasticized chitosan, the influence of acidic concentration and types on the structure and properties of thermoplastic chitosan were investigated. This work not only developed a simple method for the preparation of thermoplastic chitosan materials without using a non-volatile polyol plasticizer, but also provided a basic understanding of the melting process of plasticized chitosan with the action of an acid solution.

## 2. Materials and Methods

### 2.1. Materials

Chitosan with deacetylated degree >82.6% (8.8 wt % moisture content, *M*_w_ = 50,000 g mol^−1^, industrial grade) was bought from Shandong Laizhou Haili Biological Products Co., Ltd. (Shandong, China). Hydrochloric acid (HCl, mass fraction 36%) and acetic acid (HAc, mass fraction 99.5%) were analytical grade and bought from Sinopharm Chemical Reagent Co., Ltd. (Wuhan, China). 

### 2.2. Experiments

#### 2.2.1. Preparation of Modified Chitosan Materials

Chitosan (CS) was mixed with dilute acid aqueous solution, and the mass ratio of chitosan/acid aqueous solution was controlled to 1:1.5. Then, the mixture was processed in an internal mixer at 80 °C for 15 min. Finally, the wet solid thermoplastic chitosan materials of pale yellow were obtained. The chitosan samples were separately treated with serial aqueous solutions of different HCl concentrations 4 wt %, 6 wt %, 8 wt %, and 10 wt %, named as CH, wherein CH represents HCl-modified CS, and the number represents the mass fraction of the HCl solvent. In a similar manner, HAc aqueous solution-modified chitosan samples were designated as CA6, CA8, and CA10. The specific formulation and name of the samples are shown in Table 1. 

#### 2.2.2. Preparation of a Thermoplastic Chitosan Film

The modified chitosan materials were compression-molded using a hot press (R3202, Wuhan Qien Science and Technology Co., Ltd., Wuhan, China) equipped with a water-cooling system. The molding temperature was 95 °C and the time was 5 min, while the pressure was 40 MPa. The sheets were cut into dumbbell-like shapes according to Chinese standard GB/T1040-2006. The thickness and length of the dumbbell-like specimen and their width across the narrow section were (0.2~0.3) mm × 4 mm × 100 mm, respectively. The sheets were equilibrated at 40% RH, 60% RH, and 80% RH for 1 month before testing, respectively.

### 2.3. Characterizations

Fourier transform infrared spectroscopy (IR) was measured to analyze the structure of the composite at room temperature using an IR spectrometer (Nicolet 6700, Berkeley, Alameda County, USA). Test samples were pulverized with potassium bromide (KBr) and pressed into transparent disks for analysis. The IR spectra of all the samples were obtained with accumulation of 8 scans at a resolution of 4 cm^−1^.

The microstructure of the cross-section of the modified chitosan sheets were observed by scanning electron microscopy (SEM) (Quanta 200FEG, Hillsboro, The Netherlands) at an accelerating voltage of 5 kV. In order to obtain a cross-section, the sheet was broken with tweezers after freezing with liquid nitrogen. A thin layer of gold was sprayed on the cross-section before the test.

The composite powder was conducted with an X-ray diffractometer (XRD) (Y-2000, Dan Dong, China), the characteristic ray was CuKα (λ = 0.1542 nm), and the voltage and current were 40 KV and 30 mA, respectively. The scattering angle was 2θ = 5~40°, with a step size of 0.02°.

The light transmittance of the thermoplastic chitosan film was measured using an ultraviolet-visible absorption spectrometer (UV) (HP8453, Santa Clara, CA, USA). The thickness of the film used for the test was 0.2 mm, and the light transmittance of the film was recorded at λ = 350~800 nm.

The DSC experiments (Diamond DSC, Shang Hai, China) were conducted in a nitrogenous atmosphere. The sample was packed into an aluminum pan and sealed. Indium was used to calibrate the instrument. Prior to each test, the specimens were heated at a rate of 20 °C/min from room temperature to 110 °C to remove any thermal history, and then cooled to −50 °C at the same rate of 20 °C/min. Fusion temperatures were determined by scanning from −50 °C to 150 °C at a rate of 10 °C/min. Three duplications of this process were carried out.

Tensile testing was performed with a tensile tester (6PTS 2000S, Shen Zhen, China). The test was conducted in accordance with ASTM D882-81. The grips length was 40 mm and the strain rate was 5 mm/min_._ Tensile strength (*σ*_b_, MPa) and elongation at break (*ε*_b_, %) were obtained automatically after the tensile test. Five measurements were taken for each sample which were then averaged.

## 3. Results

### Structure

Figure 1 is the IR spectrum of the thermoplastic chitosan. It can be easily seen from Figure 1a that the peak at 1590 cm^−1^, which attributes to –NH_2_ in CS, disappeared after adding a certain amount of HCl, and correspondingly a peak at 1540 cm^−1^ emerged in CH4, CH6, CH8 and CH10 [23,24]. This demonstrates that partial –NH_2_ groups in chitosan have been protonated by the action of an acid solution. Furthermore, with the increase of HCl concentration in the acidic solution (from CH4 to CH10), the intensity of the absorption peak at 1540 cm^−1^ gradually enhances, indicating that the protonation is getting stronger. It can be seen from Figure 1b that the IR spectrum of the HAc solution modified chitosan also shows the same tendency as the HCl solution, indicating the same reaction mechanisms in the two systems.

The equations for the reactions of the HCl solution and the HAc solution with chitosan are illustrated in Figure 2, respectively. The hydrogen interactions between H^+^ in the acid solution and the –NH_2_ groups of the chitosan molecules lead to the formation of –NH^3+^ groups. At the same time, the anionic-cationic electrostatic attractions between the –NH^3+^ groups and the carboxyl groups of acetic acids, or chloridoid groups of hydrochloric acid, keep the charge balance. Although IR demonstrates that chitosan is protonated by acid solutions, it cannot distinguish the effect of the two types of acid solution in the preparing process.

Figure 3 shows the SEM of thermoplastic chitosan. Plasticizers are known to disrupt the intermolecular and intramolecular hydrogen bonds of native chitosan. In the present study, native chitosan molecules were melted or physically broken up into small fragments with the action of acid solutions. Meanwhile, the acid and water molecules penetrated into the chitosan chains and formed hydrogen bonds with chitosan molecules, which further reduced the strong intermolecular and intramolecular hydrogen bonds of chitosan. Thereby, the native chitosan chains were destroyed, with a uniform continuous phase formed. For the CH system, obvious particles could still be seen in the CH_4_, indicating that the chitosan were not completely melted. With the increase of the acid content, the surface of the chitosan sheet became flatter and the particles gradually disappeared. When the HCl content reached around to 8 wt %, all of the chitosan powders were completely melted. This is mainly because the protonation degree of chitosan gets stronger with the increase of the acid content, and the thermal melting effect is more obvious under the action of shearing force. Therefore, chitosan can be almost completely melted during processing with the HCl concentration at around 8 wt %. Compared with the CH system, the cross-section of CA sheets was obviously not as flat as the CH sheet, and a small amount of chitosan particles still remained even when the HAc concentration was 10%. Hydrochloric acid is a strong acid, its hydrogen ions can be completely ionized. While acetic acid is a weak acid, its hydrogen ions can only be partially ionized. The acid dissociation constant of HCl is higher than that of acetic acid under the same conditions, the degree of protonation between HCl and chitosan is stronger than that of HAc. Thus, the CH system was found to have a better melting effect.

The XRD pattern of the thermoplastic chitosan powder appears in Figure 4. The crystallization peaks of the pure chitosan powder at 10.6° and 20.0° respectively, correspond to the (020) and (100) reflections [25]. Specifically, the crystallization peak around 10.6° is considered to be associated with hydrogen bonding between the chitosan molecule and the water molecule, and the peak intensity is determined by the amount of moisture. When chitosan reacts with acid, a new diffraction peak appears at 22°, which is the crystal of the (110) plane [26]. It is easy to see that the intensity of the diffraction peaks of the two kinds of modified chitosan gradually decrease with the increase of the acid concentration. This phenomenon indicates that the interaction between chitosan and acid increases gradually with the increase of acid concentration, which results in the decrease of the interaction between the chitosan chains and thus affects the crystallinity of the chitosan. In addition, it can be seen from Figure 4a that when the HCl concentration was at 8%, its diffraction peak disappeared substantially, indicating that there is a strong interaction between chitosan and acid at this concentration and that some chitosan molecules may be degraded, this interaction has destroyed the original configuration of the chitosan molecule. However, the diffraction peaks of chitosan can be clearly seen in the HAc-modified material at various acid concentrations, which also proved that the interaction of chitosan with acetic acid is weaker than that of hydrochloric acid. In fact, this phenomenon can also be observed in an experiment. When the HCl concentration reached around 12 wt %, chitosan was almost completely degraded, and the material cannot be film-formed by hot compression. While the HAc modified chitosan can be film-formed even when the acetic acid is up to 90 wt. %. This result is in agreement with that observed by SEM.

Figure 5 shows the UV spectrum of thermoplastic chitosan sheets. With the increase of acid content, the light transmittance of the chitosan sheet first increased and then decreased. The light transmittance reached the maximum level when the content of HCl reached 8%. When chitosan granules were thermally melted and dissolved in water under the action of acids, its molecular chains were completely opened, and a uniform transparent film was formed after hot pressing. There were no residual chitosan particles on the surface of CH8 observed by SEM, resulting in the maximum light transmittance. However, when the concentration of HCl was continuously increased, the chitosan molecule began to degrade, and the light transmittance decreased. Since the interaction between HAc and chitosan is weaker than that of HCl, the SEM image showed that there were obvious chitosan particles on the surface of the CA sheets, and the light transmittance of the CA sheet was significantly lower than that of the CH sheet.

To study the influence of the acid solution on the thermal behavior of thermoplastic chitosan, DSC curves were obtained and are shown in Figure 6. The melting peak of chitosan with the acid concentration at around 10 wt % was not observed due to the influence of water evaporation. Since the maximal hydrochloric acid concentration cannot exceed 12 wt % in preparing thermoplastic chitosan (chitosan can completely degrade), the samples were further prepared in acetic acid solution with increased concentration. The melting temperature was not detected in native CS, attesting to the thermal degradation of chitosan before melting. The broad endothermic peak of CS at around 100 °C was also due to the influence of water. As the acid concentration increased to above 20 wt %, sharp melting endotherms appeared in all of the samples, with a melting temperature of thermoplastic chitosan around 110 °C. The above results indicate that the crystalline nature of chitosan is destructed upon heating and shearing by broking the inter/intra-molecular hydrogen bonds of chitosan. Therefore, the plasticization of CS was successful in this experiment.

Graphs of tensile test values for the thermoplastic chitosan sheet are displayed in Figure 7. Since the moisture content of the material has a great influence on its mechanical properties, we studied the effect of different humidity levels on the material properties. Figure 7a,b shows the test results of the two kinds of sheets after balancing for one month in the constant humidity system with humidity levels of 40% RH, 60% RH, and 80% RH, respectively. It can be seen from Figure 7a,b that with the increase of humidity the tensile strength of the materials decreased gradually, while the elongation at the break increased, indicating that water has a plasticizing effect on thermoplastic chitosan. The performance of both materials enhanced with the increase of acid concentration, except for sample CH10, and the performance of the CH sheet reached its maximum at 8%. In addition, the tensile strength and elongation at the break of the CA sheets were both significantly lower than those of the CH sheets at the same acid concentration. This result is inconsistent with the result reported by Marianne et al. In their study, they [27] prepared the CH sheet and the CA sheet by casting and found that the tensile strength of the CH sheet was higher than that of the CA sheet, while the elongation at the break of the two kinds of sheets exhibited the opposite order. Figure 8 shows the proposed structures of chitosan sheets via two preparation methods. For the casting method, chitosan molecules all dissolved in the acid aqueous solution. The properties of the obtained sheets are decided by the intra- and inter-molecular interactions. Due to the strong electrostatic interactions between the NH^3+^ groups and Cl^−^ (Figure 8a), the chitosan chains became rigid and their motion was restricted, resulting in a stiff and brittle film. When compared with Cl^−^, the acetate ion is larger and freer, imparting elasticity and flexibility to chitosan films (Figure 8b). For thermoplastic processing, the properties of chitosan films are primarily decided by the plasticizing effect. The more chitosan granules that are transformed into chitosan chains, the more uniform films will be formed, resulting in better film properties (Figure 8c,d).

## 4. Conclusions

In this work, thermoplastic chitosan was prepared in an internal mixer by using diluted acid solutions as a plasticizer, with the mass ratio of chitosan/acid aqueous solution at 1:1.5. The effect of the types and concentrations of diluted acid solutions on the relationship between the structure and properties of chitosan was studied. Due to being completely ionized in aqueous solution, HCl showed the stronger protonation in preparing thermoplastic chitosan. Chitosan was melted and processed when the HCl mass fraction was at 4% and was almost melted when the concentration of HCl was around 8 wt %. As a weak acid, HAc gradually improved the melt processing of chitosan when its concentration increased. The structure and properties of thermoplastic chitosan are directly affected by protonation and the performance of HCl-modified film is better than that of HAc-modified film at the same concentration. This work not only provides a basic knowledge in structure/properties of thermoplastic chitosan, but also builds a simple method for the preparation of thermoplastic chitosan materials/thermoplastic chitosan-based composites.

## Figures and Tables

**Figure 1 polymers-11-00818-f001:**
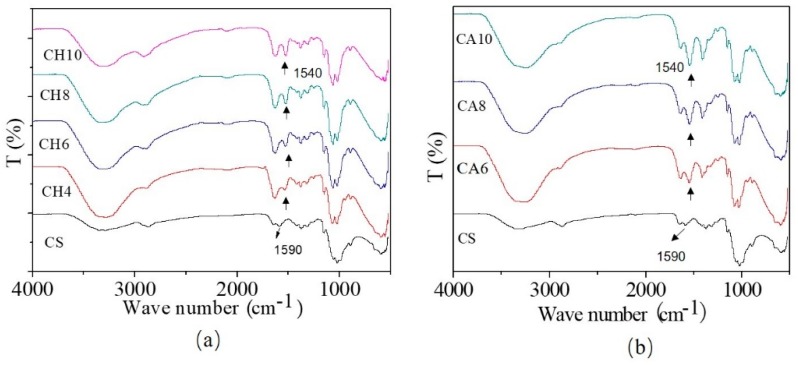
Fourier transform infrared spectroscopy (IR) spectrum of modified chitosan: (**a**) hydrochloric acid solution modified chitosan; (**b**) acetic acid solution modified chitosan.

**Figure 2 polymers-11-00818-f002:**
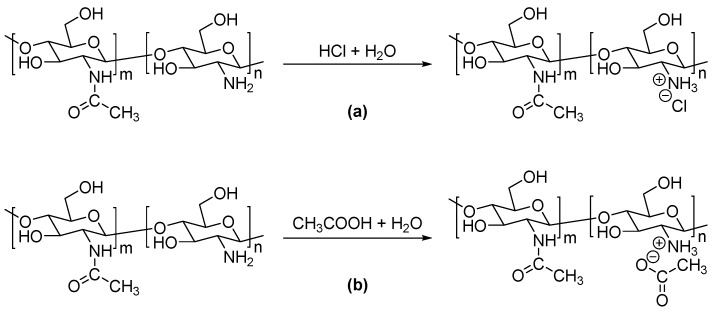
Reaction equations of chitosan with (**a**) hydrochloric acid (HCl) and (**b**) acetic acid (HAc).

**Figure 3 polymers-11-00818-f003:**
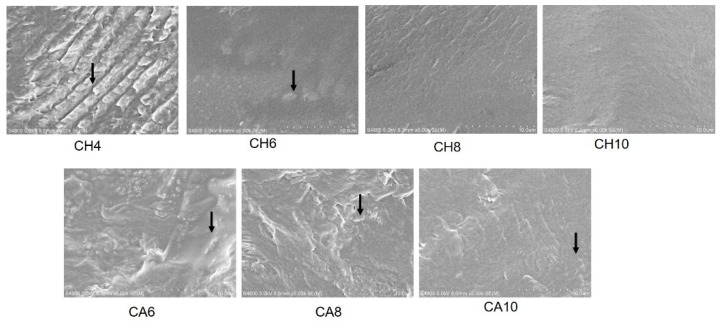
SEM of modified chitosan (5.0 KV, ×5000).

**Figure 4 polymers-11-00818-f004:**
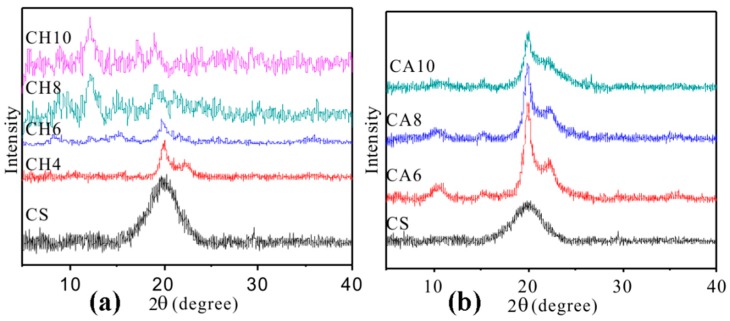
XRD pattern of modified chitosan: (**a**) hydrochloric acid solution modified chitosan; (**b**) acetic acid solution modified chitosan.

**Figure 5 polymers-11-00818-f005:**
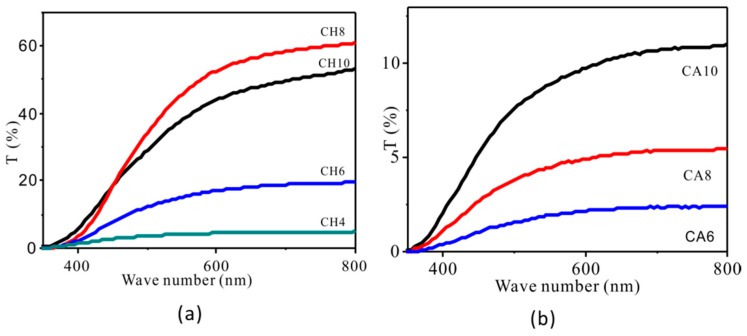
UV curve of modified chitosan: (**a**) hydrochloric acid solution modified chitosan; (**b**) acetic acid solution modified chitosan.

**Figure 6 polymers-11-00818-f006:**
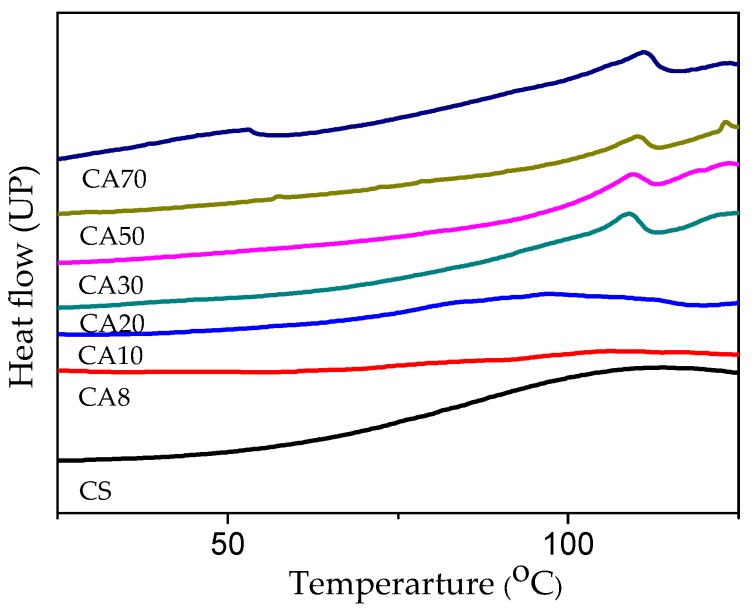
DSC curves of thermoplastic chitosan plasticized by acetic acid (CA20, CA30, CA50, and CA70 represents the acetic acid concentration in the aqueous solution used in preparing thermoplastic chitosan. The concentrations were 20 wt %, 30 wt %, 50 wt %, and 70 wt %, respectively).

**Figure 7 polymers-11-00818-f007:**
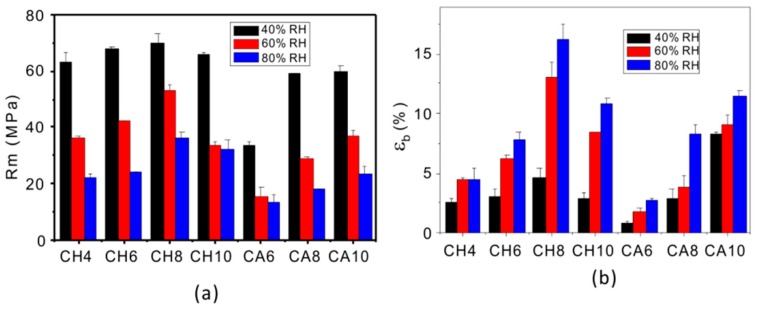
Mechanical properties of thermoplastic chitosan: (**a**) tensile strength, (**b**) elongation at break.

**Figure 8 polymers-11-00818-f008:**
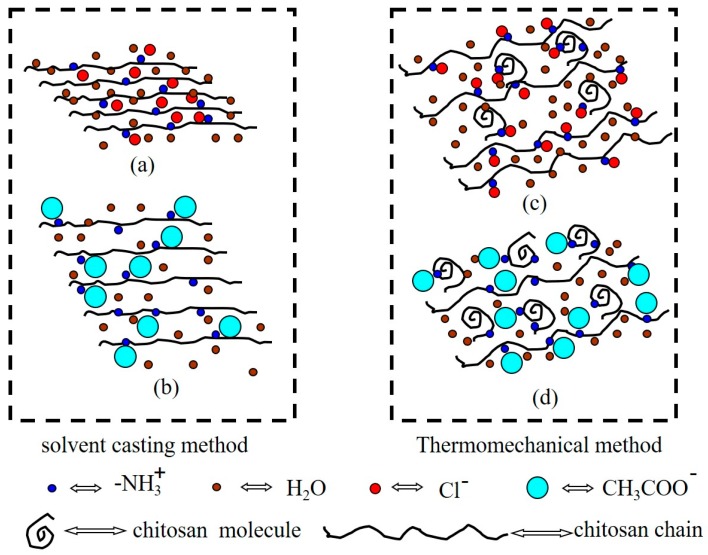
Structure diagram of modified chitosan: (**a**) preparation of chitosan film by HCl solution tape casting; (**b**) preparation of chitosan film by HAc solution tape casting; (**c**) preparation of HCl-modified chitosan film by hot pressing; (**d**) preparing an HAc-modified chitosan film by hot pressing.

**Table 1 polymers-11-00818-t001:** Specific formulation and name of the samples.

Sample Name	Chitosan (g)	36 wt % HCl (g)	HAc (g)	Water (g)	Mass Fraction of Acid (%)
CH4	30	5	--	40	4
CH6	30	7.5	--	37.5	6
CH8	30	10	--	35	8
CH10	30	12.5	--	32.5	10
CA6	30	--	2.7	42.3	6
CA8	30	--	3.6	41.4	8
CA10	30	--	4.5	40.5	10

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
