# Peer review of "Preparation, Structure and Properties of Acid Aqueous Solution Plasticized Thermoplastic Chitosan"

_polymers, 2019, doi:10.3390/polym11050818_

Reviewer 1 Report

Dear authors,

The manuscript is generally well written.

1. Please put emphasis on the novelty of this work

2. The following reference would be useful to describe some new chitosan chemistry

Carbohydrate polymers 169, 441-450

3. how were samples prepared for SEM? What voltage was used?

4. Why do XRD, it seems pointless. It is well understood that acids react with chitosan and degrade it.

5. Mw of chitosan?

6. references need to be updated with more from 2016 or newer.

8. in intro Luc Averous is not the first author of that paper, why name him specifically?

Author Response

Thanks

Reviewer 2 Report

The manuscript present the method for the preparation of thermoplastic chitosan using
 inorganic and organic acids: hydrochloric acid and acetic acid. The  optimal concentration of acids was found for obtaining needed structure and performance of the modified chitosan. The paper presents some results, but I think it is not really enough for the independent manuscript. As I see more results/discussion needed. However, I suggest this manuscript for publication if authors will be able to expand it. Also, some additional comments:

Please check affiliation, "china" has to be written from big letter

The SEM images is not very characteristic and it is difficult to make any conclusion from them. The quality should be increased, the specification should be provided in the captures to the Figure. Please, add more explanation of the results achieved by SEM.

Figure 7 capture: the double space should be removed

Author Response

Thanks

Reviewer 3 Report

The article may be considered for publication in Polymers if the authors address the following main issues:

It is not possible to refer to "granules" of chitosan without providing any evidence of the existence of such granules at the structural level (another is the case of the formulation in granules as a form of commercial presentation).

It is unacceptable to refer to "melted" granules without having carried out any thermal analysis study (by DTG, DTA or DSC) to characterize such fusion.

In my opinion, the existence of such "granules" is unlikely. What does have a basis is the existence of two forms of chitosan, perfectly defined by the authors of the article in the XRPD discussion (lines 161 to 167) and suitably schematized in Figure 7. These two forms have been accepted by the scientific community since they were reported in an article published in Thermochimica Acta, 1992, 211: 241-254: a "coiled" or "curly hair" form (determined by the existence of intramolecular hydrogen bonding) and the "uncoiled" or "sinusoidal" alternative form (when intermolecular hydrogen bonding intervenes).

Other minor issues:

The ms. would benefit from some closer proofreading. It includes linguistic errors (e.g. agreement of verbs) that should be corrected in the revised version. Taking the abstract as an example:

Line 19: define acronyms upon first usage

Line 20: … as well as a mechanical…

Line 20: … that as the acid concentration increased…

Lines 22-23: verb tenses (were optimal, was around)

Line 23: remove ‘obviously’

Line 26: …sheets [plural]… were directly affected by protonation…

Introduction: Please consider mentioning other advanced applications of chitosan-based composites (e.g., you may refer to doi: 10.1002/9781119441632.ch150, doi: 10.1186/s40643-019-0243-y, or doi: 10.1016/j.ijbiomac.2018.10.109, and references therein)

Section 2.3: Further details are needed in the description of some of the characterization methods (e.g., FTIR spectra were collected using an ATR module or KBr pellets?; what was the step size and time/step in the XRPD measurements?; etc.)

Resolution of Figure 2 needs to be improved. Please change the settings in ChemDraw (or whichever software you may have used) to export the image at a minimum of 600 dpi (preferably at 1200 dpi).

Author Response

Thanks

Round  2

Reviewer 1 Report

Dear authors,

1.the manuscript has certainly improved however in almost every single new text entry there is a mistake for example

"we only use acid aqueous solution acts as both plasticizer..."

"As shown in Figure, it is hardly to...."

There are other instances please correct.

2. it would be useful to cite recent work by tamer on using chitosan as a wound healing membrane to highlight the medical prominence of this material

MitoQ loaded chitosan-hyaluronan composite membranes for wound healing

T Tamer, et al Materials 11 (4), 569

Author Response

a

Reviewer 2 Report

In my opinion current form of manuscript is suitable for publication.

Author Response

a

Round  3

Reviewer 1 Report

I am happy with the response and effort to improve the paper